# Long-Term Changes in Adipose Tissue in the Newly Formed Bone Induced by Recombinant Human BMP-2 In Vivo

**DOI:** 10.3390/biomimetics8010033

**Published:** 2023-01-13

**Authors:** Hyouk-Keun Jee, Woo-Young Jeon, Han-Wool Kwak, Hyun Seok

**Affiliations:** 1Department of Oral and Maxillofacial Surgery, School of Dentistry, Jeonbuk National University, Jeonju 54896, Republic of Korea; 2Research Institute of Clinical Medicine of Jeonbuk National University-Biomedical Research Institute, Jeonbuk National University Hospital, Jeonju 54896, Republic of Korea

**Keywords:** rhBMP-2, adipogenesis, osteogenesis, bone scaffold, bone regeneration

## Abstract

Recombinant human bone morphogenetic protein-2 (rhBMP-2) induces osteogenesis and adipogenesis in bone scaffolds. We evaluated rhBMP-2-induced long-term changes in adipose tissue in the newly formed bone in different scaffolds forms. Bovine bone particles and blocks were grafted along with rhBMP-2 in the subperiosteal space of a rat calvarial bone, and the formation of new bone and adipose tissue were evaluated at 6 and 16 weeks after the surgery. The bone mineral density (BMD) and trabecular thickness (TbTh) of the 16w particle group were significantly higher than those of the 6w particle group (*p* = 0.018 and 0.012, respectively). The BMD and TbTh gradually increased in the particle group from weeks 6 to 16. The average adipose tissue volume (ATV) of the 6w particle group was higher than that of the 16w particle group, although the difference was not significant (*p* > 0.05), and it decreased gradually. There were no significant changes in the bone volume (BV) and BMD between the 6w and 16w block groups. Histological analysis revealed favorable new bone regeneration in all groups. Adipose tissue was formed between the bone particles and at the center in the particle and block groups, respectively. The adipose tissue space decreased, and the proportion of new bone increased in the 16w particle group compared to that in the 6w group. To summarize, in the particle group, the adipose tissue decreased in a time-dependent manner, BMD and TbTh increased, and new bone formation increased from 6 to 16 weeks. These results suggest that rhBMP-2 effectively induces new bone formation in the long term in particle bone scaffolds.

## 1. Introduction

Bone morphogenetic proteins (BMPs) were identified and reported by Dr. Marshall Urist, an orthopedic surgeon, in 1965 [1]. The BMPs are members of the transforming growth factor-beta (TGF-β) superfamily and play a key role in osteogenesis [2]. Twenty types of BMPs have been discovered, of which only BMP-2, -4, -6, -7, and -9 have osteogenic properties [3]. Initially, recombinant human BMP (rhBMP)-2 and -7 were approved for use in orthopedic applications; however, rhBMP-2 was later approved for craniofacial applications as well [1]. Recently, BMPs, especially rhBMP-2, are being used as growth factors during alveolar bone reconstruction and dental implantation [1].

An absorbable collagen sponge (ACS) was first used as an rhBMP-2 carrier in maxillary sinus augmentation and appeared to be a safe and effective substitute for new bone formation [4]. Although collagen is widely used as a carrier for rhBMP-2, water-soluble proteins are rapidly released from collagen by compression, diffusion, and degradation under physiological conditions because of its poor mechanical stability [5]. With the development of tissue engineering, many researchers have attempted to use rhBMP-2 as a bone substitute as it enhances bone healing capacity. Allogenic bone, demineralized dentin matrix, bovine bone, hydroxyapatite, and biphasic calcium phosphate (BCP) are also used with rhBMP-2, which increases new bone formation compared to the bone substitutes alone [6]. Currently, rhBMP-2 is commonly used along with bone substitutes in various clinical conditions, such as socket preservation, ridge augmentation, and guided bone regeneration. When the amount of alveolar bone is insufficient during dental implant placement, rhBMP-2 with a particulate bone scaffold provides a favorable amount of bone gain [7].

In contrast to many pro-osteogenic cytokines that exhibit anti-adipogenic properties, BMP-2 is an exception and induces the expression of PPARγ, a transcriptional regulator of adipogenesis [8]. The BMP receptor type determines whether adipogenesis or bone formation is induced; signaling through BMPR-IA exerts an adipogenic effect, whereas BMPR-IB exerts an osteogenic effect [9,10,11]. Adipogenesis that occurs during osteogenesis following BMP-2 application results in the formation of cystic-like bone voids filled with fatty bone marrow instead of the standard trabecular bone structures [12]. In an in vitro study, rhBMP-2 enhanced the adipogenic potential of stromal cells derived from the human alveolar bone in a quantity- and time-dependent manner [13]. A recent study showed that the void space volume significantly decreased two years after maxillary sinus augmentation. This clinical report concluded that void spaces can disappear clinically during long-term examinations [14]; however, it used radiological rather than histological examination.

In a previous in vivo study, the total bone and adipose tissue volumes in the grafted area were found to be higher in the bone block scaffold with rhBMP-2 than that in the bone particle scaffold with rhBMP-2. These results indicated that rhBMP-2, when used with a bone block, induced the activation of adipogenesis and osteogenesis [15]. However, there is limited information about the changes in adipose tissue inside the new bone. Therefore, this study aimed to evaluate the long-term changes in adipose tissue in newly formed bone induced by rhBMP-2 in different forms of scaffolds.

## 2. Materials and Methods

### 2.1. Animals and Materials

The animal experiments were performed with the approval of the Institutional Animal Care and Use Committee of the Jeonbuk National University Hospital, Jeonju, Republic of Korea (JBUH-IACUC-2021-33). A total of 22 Sprague Dawley rats (8-week-old; weight 200–300 g) (Samtako Biokorea, Osan, Republic of Korea) were used in this study. The rats were housed in a cage (2–3 rats per cage) under specific pathogen-free conditions and fed a standard rodent diet and water ad libitum. The rats were acclimated to the new environment for two weeks prior to surgical intervention.

Demineralized bovine bone particle and block (Bio-Oss^®^, Small granules (0.25–1 mm) and Bio-Oss Collagen^®^, Small granules Geistlich Pharma AG, Wolhusen, Switzerland) and rhBMP-2 (COWELL^®^ BMP; Cowellmedi, Busan, Republic of Korea) were used in the study. The rhBMP-2 solution was diluted in saline, and 30 µg doses were prepared. Bovine particle and block bones (0.08 g) with the rhBMP-2 solution were grafted in the subperiosteal pocket of the rat calvarium bone.

### 2.2. Surgical Intervention and Experimental Design

Surgical intervention was performed under general anesthesia induced by intramuscular injection of a mixture of Zoletil 50 (15 mg/kg; Vibac, Carros, France) and Rumpun (0.2 mL/kg; Bayer Korea, Seoul, Republic of Korea). The calvariae were shaved and disinfected with povidone-iodine. Local anesthesia was induced by injecting 2% lidocaine with epinephrine (1:100,000) into the subdermal tissue of the calvaria. All surgical interventions were performed by a single surgeon under aseptic conditions using sterile surgical instruments. A horizontal step incision was made in the posterior calvaria. Sharp subperiosteal dissection was performed, and the calvarial bone was exposed. A subperiosteal pocket was created on the surface of the parietal bone of the calvarium, and the particle and block bone (0.08 g) were grafted into the pocket along with the rhBMP-2 solution (30 µg). After grafting, the muscle and skin were closed using 3-0 Vicryl (ETHICON, Bridgewater Township, NJ, USA). Gentamycin (1 mg/kg; Kookje, Seoul, Republic of Korea) and pyrin (0.5 mL/kg; Green Cross Veterinary Products, Seoul, Republic of Korea) were injected intramuscularly thrice daily for three days. The animals were randomly divided into four groups. The 6w particle group (*n* = 5) was sacrificed six weeks after grafting the particle bone and rhBMP-2 solution, and the 16w particle group (*n* = 6) was sacrificed 16 weeks after surgery. Similarly, the 6w (*n* = 5) and 16w (*n* = 6) block groups were sacrificed 6 and 16 weeks after grafting the block bone with rhBMP-2 solution, respectively. The grafted bone materials and calvarial bone were harvested and fixed in 10% formalin. Micro-computed tomography (μ-CT) and histological analyses were performed to evaluate the changes in the newly formed bone.

### 2.3. µ-CT Analysis

The specimens were harvested, fixed in 10% formalin, and then analyzed using μ-CT at the Center for University-wide Research Facilities (CURF) at Jeonbuk National University (Jeonju-si, Republic of Korea). The specimens were scanned using a SkyScan 1076 (Bruker, Belgium) scanner with a pixel size of 35 µm. The CT scanner was set to 100 kV voltage for the X-ray tube, 100 μA current for the X-ray source, and 190 ms exposure time. The detector and X-ray source were rotated by 0.6° in 360° rotation steps. The scanned images were reconstructed using NRecon software (Bruker, Germany). The region of interest (ROI) in each sample was set as the area of grafted bone material on the surface of the calvarial bone. The bone volume (BV), bone mineral density (BMD), trabecular thickness (TbTh), trabecular separation (TbSp), and adipose tissue volume (ATV) were analyzed in the ROI for each sample using CTAn software (Bruker, Belgium). The thresholds for the new bone and adipose tissue were set in the range of 70–255 and 20–70, respectively.

### 2.4. Histological Examination

Following μ-CT analysis, the samples were decalcified in 5% nitric acid for two weeks and dehydrated in ethyl alcohol and xylene. The samples were separated through a midline sagittal suture and embedded in paraffin blocks in the sagittal plane for slicing. The paraffin blocks were sliced into sections and stained with hematoxylin and eosin. The sections showed the sagittal planes of the grafted bone material and the calvarial bone. The stained tissues were examined using an Olympus BX51 microscope (Olympus, Tokyo, Japan) and photographed using a digital camera (DP-73; Olympus, Tokyo, Japan).

### 2.5. Immunohistochemical Examination of Osteogenic Marker

The osteogenic markers bone sialoprotein (BSP) and osteocalcin were evaluated using immunohistochemistry (IHC). Anti-BSP (GTX12155; GeneTex, Irvine, CA, USA) and anti-osteocalcin (sc-365797; Santa Cruz Biotechnology) antibodies were used as the primary antibodies and detected using the Dako REAL EnVision Detection System (Dako, Glostrup, Denmark) according to the manufacturer’s protocol. Counterstaining was performed using Mayer’s hematoxylin (Sigma–Aldrich). The stained tissues were examined using an Olympus BX51 microscope (Olympus, Tokyo, Japan) and photographed using a digital camera (DP-73; Olympus, Tokyo, Japan).

### 2.6. Statistical Analysis

The BV, BMD, TbTh, TbSp, and ATV values were compared among the four independent groups. Differences among groups were evaluated using one-way analysis of variance (ANOVA; Version 23, SPSS Inc., Chicago, IL, USA) and Bonferroni’s method (post hoc test). Statistical significance was set at *p* < 0.05.

## 3. Results

### 3.1. μ-CT Analysis of BV, BMD, TbTh, and TbSp

The BV, BMD, TbTh, and TbSp values of each group were analyzed using measurements made from μ-CT images (Figure 1). The average BVs of the 6w particle, 6w block, 16w particle, and 16w block groups were 151.36 ± 30.69, 176.17 ± 71.42, 170.12 ± 24.39, and 173.96 ± 99.14 mm^3^, respectively (Figure 1a). The average BV of the 16w particle group was higher than that of the 6w particle group; however, there were no significant differences among the four groups (*p* = 0.917). The average BMDs of the 6w particle, 6w block, 16w particle, and 16w block groups were 630.36 ± 30.13, 592.11 ± 55.49, 759.74 ± 49.07, and 594.00 ± 90.58 mg/cc, respectively (Figure 1b), and there was a significant difference among four groups (*p* = 0.001). The BMD of the 16w particle group was significantly higher than that of the 6w particle (*p* = 0.018), 6w block (*p* = 0.002), and 16w block groups (*p* = 0.004).

The average TbTh values of the 6w particle, 6w block, 16w particle, and 16w block groups were 0.34 ± 0.01, 0.30 ± 0.02, 0.44 ± 0.04, and 0.40 ± 0.08 mm, respectively (Figure 1c). The average TbTh value of the 16w particle group was significantly higher than that of the 6w particle (*p* = 0.012), 6w block (*p* = 0.001), and 16w block groups (*p* = 0.038). The average TbSp values of the 6w particle, 6w block, 16w particle, and 16w block groups were 0.19 ± 0.04, 0.16 ± 0.03, 0.16 ± 0.03, and 0.19 ± 0.08 mm, respectively (Figure 1d). The average TbSp value of the 16w block group was higher than that of the other groups. No significant differences were observed among the groups (*p* = 0.531).

Three-dimensional (3D) reconstructed images observed by μ-CT analysis are shown in Figure 2. The images were cut out, and the grafted bone material was reconstructed with newly formed bone but without the calvarial bone. Both the grafted bone material (white light) and newly formed bone (green light) are shown. New bone formation and mineralization were observed on the outer surface of the bone graft material in all groups. In the sectional images, new bone formation was observed inside the bone graft material in all groups, and vacant space was also observed in the 6w block group. A marked difference appeared in the vacant or void space inside the new bone between the 6w and 16w block groups.

### 3.2. Changes in Adipose Tissue in the Newly Formed Bone

Vacant and void spaces inside the new bone are shown in the section images of μ-CT (Figure 3a–d). The 6w block group showed a vacant space inside the new bone, presumably formed by adipose tissue. Compared to the 6w block group, the 6w particle group had a void space between the bone graft material and new bone. Compared to the 6w particle and block group, the vacant and void spaces were reduced, and the volume of the graft bone appeared shrunken in the 16w group. The vacant space in the central area of the 6w block group was decreased compared to that in the 16w block group. Next, the ATV was measured using μ-CT analysis, and the time-dependent changes in ATV were compared. The results of the ATV analysis are presented in Figure 3e. The average ATV values of the 6w particle, 6w block, 16w particle, and 16w block groups were 50.30 ± 14.10, 75.20 ± 38.40, 32.19 ± 14.48, and 66.24 ± 58.93 mm^3^, respectively. The ATV of the block bone group was higher than that of the particle bone group, and it gradually decreased over time in the particle and block groups; however, there was no significant difference in ATV among the four groups (*p* = 0.264).

### 3.3. Histological Examination of the Changes in the Newly Formed Bone and Adipose Tissue

Images of the histological staining of each group are shown in Figure 4. Bone graft materials and newly formed bone were observed on the surface of the calvarial bone in all the groups. New bone formation and maturation were observed around the bone graft material, and adipose tissue was observed between the bone graft material. The adipose tissue created a vacant space with new bone formation and was distributed between the bone graft material in the particle group (Figure 4a,c), whereas it created a central vacant space inside the block bone graft material in the block group (Figure 4b,d). The particle groups exhibited lesser adipose tissue formation than the block groups at each time point. Among the particle groups, the 16w group showed less adipose tissue formation than the 6w group, indicating a decrease in the ATV over time (Figure 4a,c). In the block group, large amounts of adipose tissue were formed in the central area of the block bone graft material and were more dominant in the 6w block. The large central adipose tissue formation in the 16w block group was markedly lesser than that in the 6w block group.

### 3.4. Immunohistochemical Examination of Osteogenic Marker Expression

The expression of the osteogenic markers BSP and osteocalcin is shown in Figure 5. When rhBMP-2 was added with the two forms of bone scaffold and grafted in rat calvarial bone, favorable new bone regeneration was observed in all groups. The expression of BSP was observed in the newly formed bone and bone matrix surrounding the bone scaffold in all groups but was higher in the 6w block, 16w particle, and 16w block groups (Figure 5b–d). Osteocalcin expression was also observed in the new bone matrix of all groups, especially in the 16w particle group (Figure 5h). Between the 6w and 16w groups, BSP expression was higher in the 16w particle and block groups than in the 6w particle and block groups (Figure 5a–d). Osteocalcin expression was also higher in the 16w particle and block groups than in the 6w particle and block groups (Figure 5e–h).

## 4. Discussion

With the advancements in recombinant DNA technology, several BMPs with osteoinductive properties have been produced as recombinant human BMPs (rhBMPs) [16,17]. The bone morphogenetic proteins BMP-2 and -7 are major growth factors with osteoinductive properties, and rhBMP-2 and -7 have been developed for clinical applications [18]. The rhBMP-2 has been approved for the indications of spine fusion, tibial fracture, and craniofacial region [1,19]. Currently, rhBMP-2 is widely used in the oral and maxillofacial regions for the repair of bone defects, treatment of medication-related osteonecrosis of the jaw, and alveolar bone augmentation for implant installation [14,20,21]. In implant dentistry, rhBMP-2 is used for alveolar bone regeneration with various bone scaffolds to guide bone regeneration, maxillary sinus bone augmentation, and alveolar ridge preservation [22,23].

The bone morphogenetic protein BMP-2 is involved in the differentiation and proliferation of osteoblasts, chondrocytes, and adipocytes from undifferentiated mesenchymal stem cells [24]. It regulates the osteogenic and adipogenic differentiation of mesenchymal progenitor cells [25]. The BMP-2 also regulates the expression of peroxisome proliferator-activated receptor gamma (PPARγ), the master transcription factor of adipogenesis in mesenchymal stem cells [8]. In a previous study, rhBMP-2 produced from *Escherichia coli* (ErhBMP-2) was used with an ACS as a drug carrier and grafted in rat calvarial defects in vivo. The rhBMP-2 enhanced fatty tissue and new bone formation in a dose-dependent manner [26]. Fatty or adipose tissue formation is a clinical side effect of rhBMP-2 application during new bone formation [5]. Adipose tissue creates a void space in the newly formed bone and affects the quality of the newly formed bone induced by rhBMP-2 [27]. In another clinical study, this adipose tissue was radiographically observed as a radiolucent void space following maxillary sinus augmentation with rhBMP-2 and hydroxyapatite [28]. In another long-term clinical study, the radiographical evaluation revealed that the volume of the void space induced by rhBMP-2 gradually decreased, and osteogenesis progressed into the internal void space [14]. However, all the previous clinical studies performed radiographical evaluations, and the long-term changes in the adipose tissue have not been histologically evaluated yet.

Adipogenic differentiation is also related to the amount of rhBMP-2 used. The rhBMP-2 promotes osteogenic and adipogenic differentiation of alveolar bone-derived stromal cells and increases the adipogenic potential gradually in a dose- and time-dependent manner [13]. In our previous in vivo study, two different doses of rhBMP-2 (5 and 50 µg) were used with a bovine bone scaffold and grafted in the subperiosteal space in the rat calvarium. The high-dose rhBMP-2 group showed significantly higher osteogenic and adipogenic differentiation and tissue formation than the low-dose rhBMP-2 group [27]. The BMP-2 upregulates the osteogenic and adipogenic transcription factor, PPARγ. Consequently, favorable bone regeneration occurs with a high dose of rhBMP-2 in vivo, accompanied by the formation of adipose tissue inside the new bone. In this study, we used 30 µg of rhBMP-2 with particles and block-type bovine bones, which induced adipose tissue formation along with new bone formation. We evaluated the long-term changes in adipose tissue and compared the changes between the different forms of bone graft material.

Various scaffolds, such as collagen sponges, hydroxyapatite, and demineralized bovine bone, have been used as rhBMP-2 carriers [29,30]. In our previous in vivo study, rhBMP-2 was used with different forms of bone scaffolds, including particle- and block-type bovine bone, and adipose tissue and new bone formation were compared between the two groups [15]. When rhBMP-2 was used with bone particles, a void space was created between the space of the bone graft material and the newly formed bone; however, when rhBMP-2 was used with block bone, a void space was created in the central area of the scaffold [15]. The adipose tissue that forms inside the newly formed bone can affect the quality of the bone, and the change in this tissue needs to be evaluated for the safe clinical use of rhBMP-2. In this study, we evaluated the long-term changes in adipose tissue by histological analysis and compared the changes when rhBMP-2 was used with different forms of bone scaffolds. Changes in adipose tissue inside the new bone inevitably affect the quality of the newly formed bone. Therefore, long-term changes in adipose tissue inside the particle and block bone should be evaluated.

The particle and block bones were grafted in the subperiosteal space of rat calvaria along with 30 µg of rhBMP-2. All groups showed favorable new bone regeneration inside the bone graft scaffold 6 and 16 weeks after the surgery (Figure 4); however, there were remarkable differences in the adipose tissue formation between the particle and block bone groups. In the particle bone group, adipose tissue was formed between the space of the particle bone and newly formed bone, which appeared as a void space in μ-CT analysis; however, in the block bone group, this tissue was formed in the central area of the block bone, resulting in a large vacant space surrounded by the newly formed bone. Furthermore, the volume of adipose tissue gradually changed in a time-dependent manner, particularly in the particle bone group. The average BV of all four groups was similar, with no significant difference between the BV of the particle groups (Figure 1a, *p* = 0.917). The average BMDs of the 6w and 16w particle bone groups were 630.36 ± 30.13 and 759.74 ± 49.07 mg/cc, respectively, and the difference between them was statistically significant (Figure 1b, *p* = 0.018). The BMD increased gradually in a time-dependent manner in the particle bone group from 6 to 16 weeks. This was in contrast to the changes in the ATV. The average ATV of the 6w and 16w particle bone groups was 50.30 ± 14.10 and 32.19 ± 14.48 mm^3^, respectively, and decreased gradually with no significant difference (Figure 3e). The gradual increase in BMD in the particle bone group was attributed to the decrease in adipose tissue formation in the new bone.

Similar to the BMD results, the TbTh of the particle group gradually increased from weeks 6 to 16. The average TbTh values of the 6w and 16w particle groups were 0.34 ± 0.01 and 0.44 ± 0.04 (Figure 1c), respectively. The TbTh value of the 16w particle group was significantly higher than that of the 6w particle group (*p* = 0.012). Our results confirmed that adipose tissue in the particle bone scaffold decreased in a time-dependent manner, leading to an increase in BMD and TbTh, all of which are favorable attributes of bone quality and maturation. Our results for changes in adipose tissue and void space in vivo were consistent with the results of a previous clinical study [14]. A decrease in adipose tissue was also observed by histological examination (Figure 4).

Unlike in the particle group, there was no significant decrease in adipose tissue from 6 to 16 weeks in the block group. Therefore, there were no significant differences in BV and BMD between the 6w and 16w block groups. As in our previous study, when rhBMP-2 was used with a block-type bone scaffold, it induced osteogenesis and adipogenesis and contributed to new bone regeneration with adipose tissue formation [15]. In contrast to its formation in the particle bone scaffold, adipose tissue was formed in the central area of the block bone scaffold and created a large vacant space. In this study, we compared the long-term changes in adipose tissue and hypothesized that the vacant space in the block group would decrease and osteogenesis would progress inside the vacant space over time, leading to an increase in bone quantity and quality. However, contrary to our hypothesis, the μ-CT analysis and histologic examination revealed that the adipose tissue and vacant space did not change over 16 weeks. Therefore, the BV and BMD values in the block group did not change over 16 weeks (Figure 2 and Figure 4). The cause for the formation of the central vacant space in the block remains to be elucidated; this phenomenon is possibly related to the rigidity of the block bone scaffold. A plausible explanation may be that the particle bone is not attached to each granule and is separated such that the new bone and adipose tissue can grow in the space of each particle bone, creating a void space between the particles. The block-type bone scaffold used in our study was composed of approximately 90% deproteinized bovine bone and 10% of porcine collagen [31]. Porcine collagen acts as a binder for bovine bone [31]. Block bone has outstanding spatial stability and is widely used as a bone graft material for tooth extraction and socket preservation [32,33]. The stability of block-type bones affects the formation of adipose tissue in the central area. On histological examination, new bone and residual bone material were found surrounding the adipose tissue, thus creating a central vacant space (Figure 3 and Figure 4).

Favorable new bone regeneration was histologically observed in all groups, and rhBMP-2 successfully induced new bone regeneration when added to the bone scaffold (Figure 4). Adipogenesis was also observed in all groups along with new bone regeneration. The adipose tissue was located in the space between the bone particles in the particle groups (Figure 4a,c) and the center of the bone block in the block groups (Figure 4b,d). Compared to the 6w particle group, the space of adipose tissue decreased, and the proportion of new bone increased in the 16w particle group (Figure 4a,c); however, this pattern was not obvious in the block group (Figure 4b,d). Bone sialoprotein (BSP), a non-collagenous protein, is a crucial component of the bone extracellular matrix [34]. It is a key component of mineralized bone tissue, predominantly observed in newly formed bone matrix induced by rhBMP-2 [27]. Bone sialoprotein was highly expressed in the new bone matrix in all groups and detectable in the areas surrounding the bone scaffold and adipose tissue, especially in the 6w block, 16w particle, and 16w block groups. Osteocalcin, a non-collagen protein hormone produced by osteoblasts and found in bone and dentin, is often used as a marker of bone formation [35,36]. Osteocalcin was also expressed in the newly formed bone matrix in all groups. Compared to the 6w particle and block groups, BSP and osteocalcin expression tended to be higher in the 16w particle and block groups. Therefore, bone maturation and mineralization progressed in a time-dependent manner as expected.

In a previous clinical study, bone augmentation with rhBMP-2 and particle bone material initially created void spaces inside the graft material; however, it gradually disappeared, and osteogenesis progressed in the void space 24 months after the surgery [14]. In our study, the particle groups showed a void space between the grafted particle bone, which gradually decreased in the 16w groups. This led to a time-dependent increase in BMD and TbTh in the particle groups. Bone grafting surgery for implant installation has been widely performed in patients with alveolar BV deficiency [37]. Popularly used with bone grafting material, rhBMP-2 successfully induces new bone regeneration in guided bone regeneration (GBR), maxillary sinus augmentation, and bone defects after cyst enucleation [23,38]. When used with hydroxyapatite granules, rhBMP-2 induces significantly higher bone formation than bovine bone xenografts alone in maxillary sinus augmentation [6]. However, when a high dose of rhBMP-2 is grafted with BCP in the maxillary sinus, it induces radiolucent void artifacts with dispersed particles, inhibiting new bone regeneration in vivo [39]. The rhBMP-2 has both osteogenic and adipogenic differentiation potential, and the void space formed after maxillary sinus augmentation with rhBMP-2 contained adipose and fatty marrow-like tissue [28]. This adipose tissue inhibits new bone regeneration in the early healing stage following sinus augmentation [39]. Long-term changes in void space induced by rhBMP-2 have been reported. The void space gradually disappeared as osteogenesis progressed over 24 months following maxillary sinus augmentation; however, these studies were only performed using radiological analysis and not histological examination [14]. Contrary to previous studies, we evaluated the long-term changes in adipose tissue using μ-CT, histological, and IHC analyses and confirmed that the void space and adipose tissue in the particle group decreased in a time-dependent manner and that new bone formation spontaneously increased in the long-term evaluation.

The use of rhBMP-2 with particle bone and block bone scaffolds induces both new bone and adipose tissue, which appears as a void or vacant space. In contrast to the observation in rhBMP-2 with the block bone group, the void space in the particle bone group decreased and was occupied by new bone, leading to an increase in BMD and TbTh in the long-term evaluation. Although rhBMP-2 induced both osteogenesis and adipogenesis when used with particle bone scaffold, the adipose tissue decreased in a time-dependent manner, and new bone formation progressed into the void space. Therefore, rhBMP-2 was more effective in inducing new bone formation when using a particle bone scaffold in the long term.

## Figures and Tables

**Figure 1 biomimetics-08-00033-f001:**
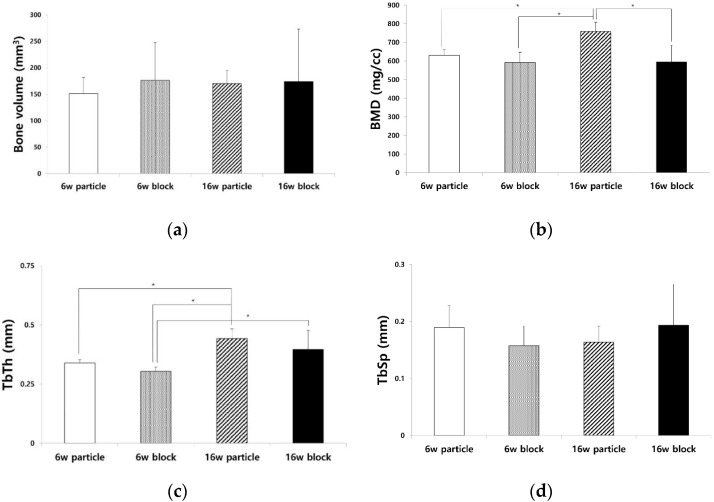
Results of μ-CT analysis. (**a**) Bone volume (BV), (**b**) bone mineral density (BMD), (**c**) trabecular thickness (TbTh), and (**d**) trabecular spacing (TbSp) of 6w particle, 6w block, 16w particle, and 16w block groups. BMD and TbTh were significantly different among the four groups (*p* = 0.001, * *p* < 0.05).

**Figure 2 biomimetics-08-00033-f002:**
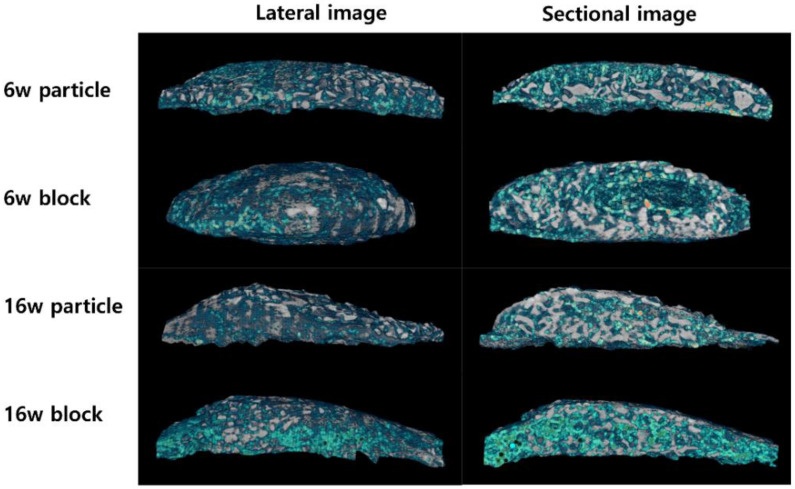
Three-dimensional (3D) reconstruction images from micro-computed tomography (μ-CT) analysis of 6w particle, 6w block, 16w particle, and 16w block groups showing the newly formed bone (green light) and grafted bone scaffold (white light).

**Figure 3 biomimetics-08-00033-f003:**
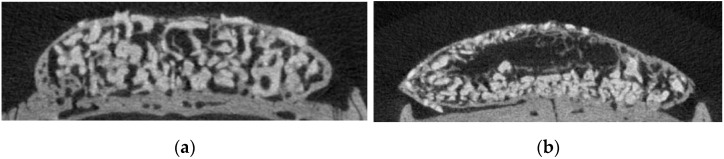
μ-CT images and volume of adipose tissue (ATV) in 6w particle, 6w block, 16w particle, and 16w block groups. (**a**,**b**) Void space formation in 6w and 16w particle groups, (**c**,**d**) vacant space in the central area of block bone graft material in 6w and 16w block groups. (**e**) There was no significant difference in the ATV among the four groups (*p* = 0.264).

**Figure 4 biomimetics-08-00033-f004:**
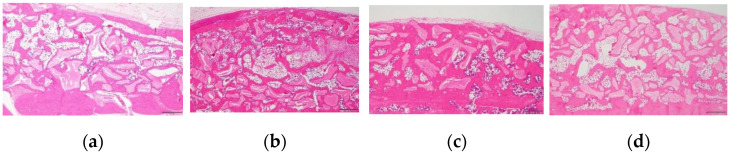
Histological images (hematoxylin and eosin staining) of each group. (**a**,**e**) 6w particle, (**b**,**f**) 6w block, (**c**,**g**) 16w particle, and (**d**,**h**) 16w block groups. New bone and adipose tissue formation were observed in the particle groups. Large amounts of adipose tissue formation are visible in the central area of bone graft material in the block groups. Panels (**e**–**h**) represent higher magnification of Panels (**a**–**d**), respectively. Black asterisks = bone scaffold ((**a**–**d**); original magnification 40×, bar = 500 µm, (**e**–**h**); original magnification 100×, bar = 200 µm).

**Figure 5 biomimetics-08-00033-f005:**
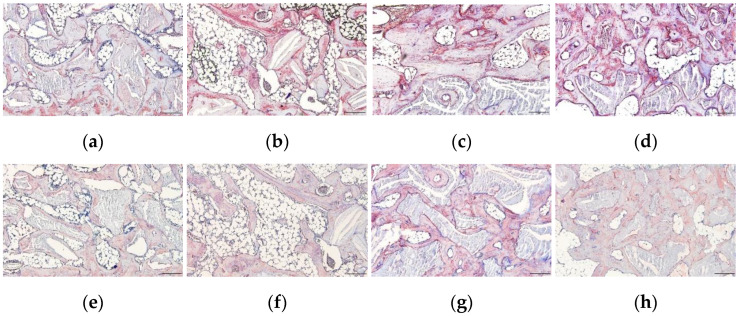
Immunohistochemical staining of bone sialoprotein (BSP) and osteocalcin in each group. (BSP; **a**–**d**) and osteocalcin (**e**–**h**) after surgical intervention. (**a**,**e**) 6w particle, (**b**,**f**) 6w block, (**c**,**g**) 16w particle, and (**d**,**h**) 16w block groups. High expression levels of BSP and osteocalcin were observed in the newly formed bone (original magnification 100×, bar = 200 µm).

## Data Availability

Not applicable.

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
