# Peer review of "Long-Term Changes in Adipose Tissue in the Newly Formed Bone Induced by Recombinant Human BMP-2 In Vivo"

_biomimetics, 2023, doi:10.3390/biomimetics8010033_

Round 1

Reviewer 1 Report

Thank you for your novel manuscript. 

overall good, but reviewer have a few question

Q1. Author mentioned "The 6w block group showed a vacant space inside the new bone, presumably formed by adipose tissue." 

Please explain in detail and mention the limitation

And ATV(adipose tissue volume) in uCT, not MRI how do you measure the adipose tissue volume?

Q2. Author use Bio-oss as particle and Bio-oss collagen as block, describe about in detail(ex. collagen contents of block, particle, pore size) 

Author Response

Q1. When we analyzed the micro-CT, we set differently for newly formed bone analysis and adipose tissue analysis. The threshols for the new bone and adipose tissue were set in the range of 70–255 and 20–70. And we referred to the histological images for set that thresholds.

Q2. We used Bio-oss and Bio-oss collagen with small granules which size is 0.25-1mm. We revise the material and methods adding this information. And the content of Bio-oss is 90% deproteinized bovine bone and 10% of porcine collagen, it was already described in the 7th paragraph in discussion.

Reviewer 2 Report

The research presents a high-quality elaboration and presentation. I congratulate the authors for their work.

I have to admit that I made a very concise evaluation of the paper, and I'm glad you gave me this opportunity to better share my thoughts.

I have been working with bone repair using biomaterials and stem cells, but issues related to the adipose tissue distributed throughout bone marrow have never been my focus before, although many alterations were quite evident.

I always call the attention of graduate students who are interested in orthopedics about how the surgical technique they use reflects on a molecular process, which is to provide the necessary conditions for the precursor mesenchymal cells that reach the fracture site to receive the correct niche information, and showing how these early repair events are common to other tissue types. In some, it's okay if those cells produce a scar, but not in bone.

With that in mind, I made this very positive assessment of the work that you entrusted me to evaluate, as it shows, through a simple and accessible methodology, a little more than what we need to observe in some cases, opening up many possibilities from there.

Author Response

Thanks for your kind advice and encourage.
